# Diagnostic Performance Evaluation of the Novel Index Combining Urinary Cotinine and 4-(Methylnitrosamino)-1-(3-pyridyl)-1-butanol in Smoking Status Verification and Usefulness for Trend Monitoring of Tobacco Smoking Exposure

**DOI:** 10.3390/ijerph191912147

**Published:** 2022-09-25

**Authors:** Hyun-Seung Lee

**Affiliations:** Department of Laboratory Medicine, School of Medicine, Wonkwang University, 895 Muwang-ro, Iksan-si 54538, Jeollabuk-do, Korea; ctmasaru@gmail.com; Tel.: +82-10-2631-4590; Fax: +82-63-842-3786

**Keywords:** cotinine, NNAL, biomarker, e-cigarette, secondhand smoke, smoking status verification

## Abstract

During the last decade in Korea, urinary cotinine concentrations in non-current smokers have decreased, making it difficult to distinguish secondhand smoke (SHS) exposure from nonsmokers because of overlapping values between non-current smokers with and without SHS exposure. Additionally, the importance of smoking status verification to avoid misclassification is increasing with the increased use of e-cigarettes. We developed a novel index combining urinary cotinine and 4-(methylnitrosamino)-1-(3-pyridyl)-1-butanol (NNAL) and evaluated its diagnostic performance for the classification of smoking status using the KNHANES VII dataset. A total of 10,116 and 5575 Korean participants aged >19 years with measured urinary cotinine concentrations were enrolled in a training set and validation set, respectively. When using 4.0 as the cutoff value for distinguishing current smokers from non-current smokers, urinary cotinine∙NNAL showed a better diagnostic performance than urinary cotinine or urinary NNAL. Among e-cigarette users, urinary cotinine∙NNAL showed more accurate classification rates than urinary NNAL. Furthermore, urinary cotinine∙NNAL had measurable values in non-current smokers, whereas urinary cotinine had unmeasurable values in one-fourth of all participants. This study shows that urinary cotinine∙NNAL might be a useful biomarker for smoking status verification and trend monitoring of tobacco smoking exposure with increased use of e-cigarettes.

## 1. Introduction

Tobacco smoking exposure, which is a common public health issue and a cause of preventable morbidity and mortality worldwide [1], is associated with the risk of asthma, respiratory tract infection, and various cancers [2,3,4,5]. Accurate smoking status verification and trend monitoring of tobacco exposure could play a vital role in public health policy and tobacco exposure regulations. Measurement of biomarkers constitutes the most widely used objective method for smoking status verification [6].

As a biomarker of tobacco smoking exposure, cotinine has been studied and used in population surveys to establish public policies regarding secondhand smoke (SHS) exposure [7]. Cotinine is a major metabolite of nicotine that can be measured in serum, urine, saliva, and hair [8]. Urinary cotinine is a noninvasive biomarker, and its diagnostic performance is comparable to that of serum cotinine [9,10]. Thus, the Korean National Health and Nutrition Examination Survey (KNHANES) has used urinary cotinine as a biomarker for many years to validate reported smoking status and monitor population exposure to tobacco over time [10]. The average half-life of plasma cotinine is 16–18 h [11], and that of urinary cotinine is approximately 20 h in adults [12]. The half-life of cotinine is longer than that of nicotine or carbon monoxide [13]; however, cotinine might not be ideal for validating smoking cessation for several days. Moreover, cotinine might be less sensitive than a biomarker with a longer half-life for monitoring SHS exposure.

4-(Methylnitrosamino)-1-(3-pyridyl)-1-butanol (NNAL) is a metabolite of 4-(methylnitrosamino)-1-(3-pyridyl)-1-butanone (NNK), which is a tobacco-specific nitrosamine (TSNA) [14]. Urinary NNAL has a long half-life of 10–16 days [15]; therefore, it would be more sensitive to detect tobacco smoke exposure or SHS exposure over a much longer period than cotinine [16]. A previous study showed that 94% of adolescents had concentrations higher than the lower limit of quantitation (LoQ) of NNAL compared to 87% for cotinine [17]. However, NNAL might be less sensitive for the verification of smoking status with the use of an electronic cigarette (e-cigarette) exclusively or nicotine replacement products exclusively. With the increased use of e-cigarettes [13], NNAL might result in a misclassification of smoking status and an underestimation of trend monitoring for tobacco smoke exposure or SHS exposure [17].

The development of a novel index based on a combination of multiple biomarkers is useful for making medical decisions in clinical practice [18]. The combination of urinary cotinine and NNAL might be advantageous for smoking status verification and trend monitoring of tobacco smoke exposure or SHS exposure. A few studies have reported that the ratio of urinary NNAL to cotinine (NNAL/cotinine) is higher in people exposed to SHS than in active smokers [15,19]. In the current study, a novel index combining urinary cotinine and NNAL was developed, and the diagnostic performance was evaluated for the classification of smoking status using the KNHANES VII dataset (conducted between 2016 and 2018), which contained both urinary cotinine and NNAL data. In addition, the usefulness of the novel index was assessed for the verification of smoking status with e-cigarette use and to detect SHS exposure in non-smokers.

## 2. Materials and Methods

### 2.1. Study Participants

A schematic flow chart of the study design and exclusion criteria is shown in Figure 1. Of the total of 24,269 participants who participated in the 2016, 2017, and 2018 KNHANES surveys, which were conducted by the Korea Centers for Disease Control and Prevention, 4880 participants aged <19 years were excluded. Of the 19,389 adult participants, 3498 were excluded for whom urinary cotinine concentrations were not measured. Finally, 15,891 Korean participants aged ≥19 years with measured urinary cotinine concentrations were enrolled. This study was approved by the institutional review board of Wonkwang University Hospital (IRB file no. 2022–07–010). A waiver of consent was obtained, given the retrospective nature of the study, which aimed to discover a novel index combining multiple biomarkers using a public dataset.

### 2.2. Self-Report for Smoking Status

In the current study, definitions of self-reported smoking status were the same as those used in a previous study [20] for establishing urinary cotinine-based optimal cut-off values for smoking status classification. Briefly, no response to smoking status was defined as a participant who did not participate in the self-report of smoking status or did not answer the self-report. Current smokers were defined as participants who reported a history of smoking 100 cigarettes in their lifetime and currently smoking cigarettes. An active smoker was defined as a participant who reported, “Yes, I smoke at least one cigarette a day” among current smokers. Non-daily smokers were defined as participants who reported “Yes, I smoke, but not every day” among current smokers. Ex-smokers were defined as participants who reported, “No, I don’t smoke but smoked past” among non-current smokers. Non-smokers were defined as participants who did not meet the current smoker or ex-smoker definition for self-reported smoking status. Nicotine replacement product (NRP) users or e-cigarette users were defined as participants who reported a history of NRP or e-cigarette use within 30 days.

### 2.3. Self-Report for SHS Exposure

In the current study, the definition of self-report for SHS exposure was the same as that in a previous study [20]. Briefly, SHS exposure was defined as a participant who reported a history of SHS exposure at home or in the workplace among non-current smokers. An unclear response to SHS exposure was defined as a participant who was not included in SHS exposure and who reported an unclear history of SHS exposure, such as no response, unknown response, and non-defined response, at home or in the workplace, among non-current smokers. Non-SHS exposure is defined as a participant who definitively reported no history of SHS exposure at home and in the workplace among non-current smokers.

### 2.4. Measurement of Urinary Cotinine and NNAL

A high-performance liquid chromatography-tandem mass spectrometry (HPLC-MS/MS) system comprising a 1100 HPLC system (Agilent, Santa Clara, CA, USA) and API 4000 (AB Sciex, Redwood City, CA, USA) was used to measure the urinary cotinine levels. An HPLC-MS/MS system comprising a 1200 HPLC system (Agilent, Santa Clara, CA, USA) and a Triple Quadrupole 5500 (AB Sciex, Redwood City, CA, USA) was used to measure urinary NNAL concentrations. The limits of detection (LoD) of urinary cotinine and NNAL were 0.2740 µg/L and 0.1006 ng/L, respectively.

### 2.5. Statistical Analyses

In the current study, the measured values equal to or lower than the LoD were converted to LoD values. Histograms of urinary cotinine and NNAL are expressed as a log scale and percentage frequency. Data with a normal distribution are expressed as mean ± standard deviation (SD) while skewed data are expressed as median (interquartile range (IQR)). The area under the curve (AUC)–receiver operating characteristic (ROC) curve analysis with the maximum value of Youden’s index was used to establish optimal cut-off values of urinary cotinine, urinary NNAL, urinary NNAL/cotinine, and the novel index combined with urinary cotinine and NNAL for smoking status classification. Fisher’s exact test or chi-square test was used to analyze categorical data. The Mann–Whitney U test or Kruskal–Wallis test was used to analyze continuous data. Statistical Package for the Social Sciences version 25.0 (IBM Corporation, Armonk, NY, USA), MedCalc Statistical Software version 19.2.6 (MedCalc Software, Ostend, Belgium), and GraphPad Prism version 9.1.2. (GraphPad Software, La Jolla, CA, USA) were used for statistical analyses and graphs. Statistical significance was set at *p* < 0.05.

## 3. Results

### 3.1. Characteristics of Study Participants

The 2016 and 2017 datasets were used as a training set, and those of 2018 were used as a validation set. The characteristics of the participants are listed in Table 1. The total number of enrolled participants in the training and validation sets was 10,116 and 5775, respectively. The percentage of females was 53.5% (5397/10,116) and 55.0% (3179/5775) in each set. The mean age of participants was 51.1 ± 16.5 and 49.1 ± 16.8 years, respectively. Among them, 37.9% (3841/10,116) and 37.2% (2148/5775) had NNAL values. The mean values of urinary cotinine were 286.8 ± 648.8 and 300.0 ± 682.2 µg/L, and those of urinary NNAL were 52.5 ± 131.7 and 45.9 ± 121.2 ng/L, respectively. Participants with equal to or less than LoD of urinary cotinine were 12.1% (1225/10,116) and 21.1% (1216/5775), and those of urinary NNAL were 0.2% (9/3841) and 0.6% (13/2148), respectively.

### 3.2. Results of the Self-Reported Smoking Status

The results of the self-reported smoking status are shown in Table 1. The proportion of participants who responded to the self-report of smoking status in the training and validation sets was 98.9% (10,002/10,116) and 99.4% (5739/5775), respectively. Among them, current smokers were 20.7% (2075/10,002) and 18.2% (1047/5739) and non-current smokers were 79.3% (7927/10,002) and 81.8% (4692/5739), respectively. Among current smokers, 86.6% (1796/2075) and 86.5% (906/1047) were daily smokers, and 13.4% (279/2075) and 13.5% (141/1047) were non-daily smokers, respectively. Among non-current smokers, ex-smokers accounted for 26.8% (2124/7927) and 26.9% (1264/4692) while non-smokers were 73.2% (5803/7927) and 73.1% (3428/4692), respectively. NRP users accounted for 0.8% (80/10,002, 44/5739) while e-cigarette users accounted for 1.8% (182/10,002) and 2.8% (163/5739), respectively.

### 3.3. Results of the Self-Reported SHS Exposure

The proportion of participants with SHS exposure in the validation set was 6.8% (320/4692) while that of non-SHS exposure participants was 90.3% (4236/4692). The percentage of participants who submitted unclear self-reports of SHS exposure was 2.9% (136/4692). Of the participants with SHS exposure, 10.3% (33/320) of urinary cotinine and 0% (0/290) of urinary NNAL were equal to or less than the LoD, respectively.

### 3.4. Scatter Plot of Urinary Cotinine and NNAL in the Training Set

The scatter plot of log-transformed urinary cotinine and NNAL levels in the training set is shown in Figure 2A. Two major clusters and two minor clusters were observed based on visual inspection of the scatter plot. One of the major clusters was located at >100 µg/L of urinary cotinine and >10 ng/L of urinary NNAL. This cluster consisted mainly of daily smokers, non-daily smokers, and ex-smokers. The other major cluster was located at less than 10 µg/L of urinary cotinine and less than 10 ng/L of urinary NNAL, and mainly consisted of non-smokers and ex-smokers. Two minor clusters were mixed: current and non-current smokers. One minor cluster was located at >10 µg/L of urinary cotinine and less than 10 ng/L of urinary NNAL while the other was located at <100 µg/L of urinary cotinine and >10 ng/L of urinary NNAL.

### 3.5. Scatter Plot of Urinary Cotinine and a Novel Index Multiplying Urinary Cotinine and NNAL in the Training Set

A novel index combining urinary cotinine and NNAL was calculated as follows:$$\mathrm{Urinary}\;\mathrm{cotinine}\cdot \mathrm{NNAL}={\mathrm{Log}}_{10}\left(\mathrm{urinary}\;\mathrm{cotinine}\times \mathrm{urinary}\;\mathrm{NNAL}\right)+2$$

The scatter plot of log-transformed urinary cotinine and NNAL levels in the training set is shown in Figure 2B. The participants with values less than two of urinary cotinine∙NNAL were mainly non-smokers while the participants with values greater than four were mainly current-smokers and ex-smokers. The range between two and four urinary cotinine∙NNAL mainly consisted of non-smokers and ex-smokers in the scatter plot.

### 3.6. Distributions of Urinary Biomarkers from All Participants

The distribution of four urinary biomarkers, consisting of urinary cotinine, urinary NNAL, urinary cotinine∙NNAL, and urinary NNAL/cotinine from all participants in the training set, is illustrated in Figure 3. On visual inspection of the histogram, urinary cotinine and urinary cotinine∙NNAL had a relatively wider interval between current smokers and non-current smokers compared to urinary NNAL and urinary NNAL/cotinine. In contrast, urinary NNAL had measurable values for non-current smokers, whereas urinary cotinine had values less than LoD in about 25% of all participants. When the values of urinary cotinine or urinary NNAL with equal to or lower than the LoD were converted to LoD values, most of urinary cotinine∙NNAL and urinary NNAL/cotinine were higher than the minimum values, which were calculated with each LoD value. Collectively, the distribution analysis suggests that urinary cotinine∙NNAL might be more appropriate than other urinary biomarkers for smoking status verification and trend monitoring of SHS exposure.

### 3.7. Established Optimal Cut-Off Values of Urinary Biomarkers Using the Training Set and Diagnostic Performance Evaluation for Smoking Status Classification

Using the training set, the established optimal cutoff values of the four urinary biomarkers and their diagnostic performance for smoking status classification are described in Table 2. Each cutoff value was established to distinguish current smokers from non-current smokers. The optimal cut-off values for distinguishing current smokers from non-current smokers in the training set were 26.9 µg/L (95% confidence interval, 9.3–33.9 µg/L, urinary cotinine), 12.3 ng/L (12.0–18.6 ng/L, urinary NNAL), 4.0 (3.9–4.6, urinary cotinine∙NNAL), and 0.6 mg/g (0.6–0.8 mg/g, urinary NNAL/cotinine), respectively. The diagnostic performance for each optimal cutoff varied from 94.70–98.45% for sensitivity and 88.15–95.33% for specificity in the training set. When applying the established cut-off in the validation set, the diagnostic performance for each biomarker varied from 96.20–98.66% for sensitivity and 89.35–96.01% for specificity According to AUC values, urinary cotinine∙NNAL showed the best diagnostic performance for distinguishing current smokers from non-current smokers (AUC = 0.985 and 0.987 in the training and validation sets, respectively), whereas urinary NNAL/cotinine showed the worst diagnostic performance (AUC = 0.937 and 0.942, respectively).

### 3.8. Comparison of the Diagnostic Performance of Urinary Biomarkers in E-Cigarette Users and NRP Users

A comparison of the diagnostic performance of the four urinary biomarkers in e-cigarette and NRP users is presented in Table 3. The median values of urinary cotinine and urinary cotinine∙NNAL were similar between e-cigarette users and current smokers, whereas those of urinary NNAL and urinary NNAL/cotinine were lower in e-cigarette users compared to current smokers. Urinary NNAL showed 10.1% (8/77 in the training set) and 21.4% (15/70 in the validation set) misclassification rates for e-cigarette users, whereas other biomarkers showed misclassification rates of 2.5–6.3%. According to the misclassification rates for e-cigarette users, urinary cotinine and urinary cotinine∙NNAL showed the most accurate classification rates in the training and validation sets, whereas urinary NNAL showed the worst misclassification rates.

On the other hand, each biomarker showed 81.8%–90.0% misclassification rates for NRP users in the training set. Due to the small sample size, the diagnostic performance of the four urinary biomarkers in NRP users could not be evaluated in the current study.

### 3.9. Comparison of Diagnostic Performance in Non-Current Smokers according to SHS Exposure

A comparison of the diagnostic performance of SHS exposure in non-current smokers in the validation set is presented in Table 4. The median values of each biomarker were higher in non-current smokers with SHS exposure than in non-current smokers without SHS exposure. However, urinary cotinine was unmeasurable in 10.2% (33/320) of participants with SHS exposure and in 3.0% (4/136) of participants with unclear SHS exposure, whereas urinary NNAL was perfectly measurable. Compared to urinary NNAL, urinary cotinine showed a methodological limitation for trend monitoring of SHS exposure in the current study.

## 4. Discussion

My previous study showed that decreased SHS exposure results in decreased optimal cut-off values for distinguishing current smokers from non-current smokers [20]. During the last decade in Korea, the median value of urinary cotinine in non-current smokers decreased from 5.86 to 0.48 µg/L, making it difficult to distinguish SHS exposure from nonsmokers due to overlapped values between non-current smokers with and without SHS exposure. In addition, this study showed the importance of smoking status verification to avoid misclassification in the increased use of e-cigarette settings. These findings suggest the need for a novel biomarker that is more sensitive for distinguishing SHS exposure from nonsmokers and provides accurate classification of smoking status, regardless of e-cigarette use.

Urinary NNAL is an alternative biomarker for smoking status verification that might be more suitable for the detection of SHS exposure over a much longer period than urinary cotinine. However, the concentration of NNK in tobacco products, a precursor of NNAL, varies according to the e-cigarette brand [13]. NNK is found only in tobacco products and not in green tobacco plants because it is formed from tobacco-specific alkaloids during curing and processing [13]. According to a previous study on NNK concentrations in different e-liquids, 33% (4/12) had unmeasurable amounts of NNK [21]. Therefore, exclusive e-cigarette users and NRP-exclusive users might have significantly lower NNAL concentrations than combustible cigarette exclusive users or dual combustible cigarette-e-cigarette users [17]. In the current study, urinary NNAL showed higher misclassification rates than the other biomarkers in e-cigarette users, and the misclassification rates of urinary NNAL increased with an increase in e-cigarette use.

Urinary cotinine∙NNAL, a novel index combining urinary cotinine and NNAL, was designed to merge the advantages and reduce the disadvantages of the two tobacco-related biomarkers. According to the scatter plot of log-transformed urinary cotinine and NNAL, urinary NNAL/cotinine could be helpful in distinguishing one of the two minor clusters from non-current smokers, which had decreased urinary cotinine concentrations and elevated urinary NNAL concentrations, whereas it could not distinguish the other, which had elevated urinary cotinine concentrations and decreased urinary NNAL concentrations. In contrast, urinary cotinine∙NNAL could be useful in distinguishing between the two minor clusters from non-current smokers. When using 4.0 as the cutoff value for distinguishing current smokers from non-current smokers, urinary cotinine∙NNAL showed a better diagnostic performance than urinary cotinine or urinary NNAL. Furthermore, urinary cotinine∙NNAL showed a diagnostic performance similar to that of urinary cotinine in e-cigarette users. Urinary cotinine∙NNAL and urinary cotinine showed more accurate classification rates than urinary NNAL in the current study. In addition, urinary cotinine∙NNAL showed a distribution pattern similar to that of urinary NNAL in non-current smokers (Figure 4). According to the distribution analysis, log-transformed urinary cotinine∙NNAL and urinary NNAL had almost measurable values for non-current smokers, whereas urinary cotinine had unmeasurable values in one-fourth of the participants. In addition, urinary cotinine∙NNAL was perfectly measurable among participants with SHS exposure whereas urinary cotinine was not. Taken together, urinary cotinine∙NNAL has a better diagnostic performance than urinary cotinine or urinary NNAL, and might be useful for trend monitoring of tobacco smoking exposure and SHS exposure due to its wider dynamic range compared to urine cotinine.

The current study has several limitations. First, e-cigarette exclusive users or dual combustible cigarette-e-cigarette users could not be distinguished due to limited questionnaire information. Next, the diagnostic performance of each urinary biomarker in NRP users could not evaluated due to the small sample size in the current study. Further study is needed to solve these limitations.

## 5. Conclusions

Collectively, this study showed that urinary cotinine∙NNAL, a novel index combining urinary cotinine and NNAL, might be a useful biomarker for smoking status verification and trend monitoring of tobacco smoking exposure with increased use of e-cigarettes.

## Figures and Tables

**Figure 1 ijerph-19-12147-f001:**
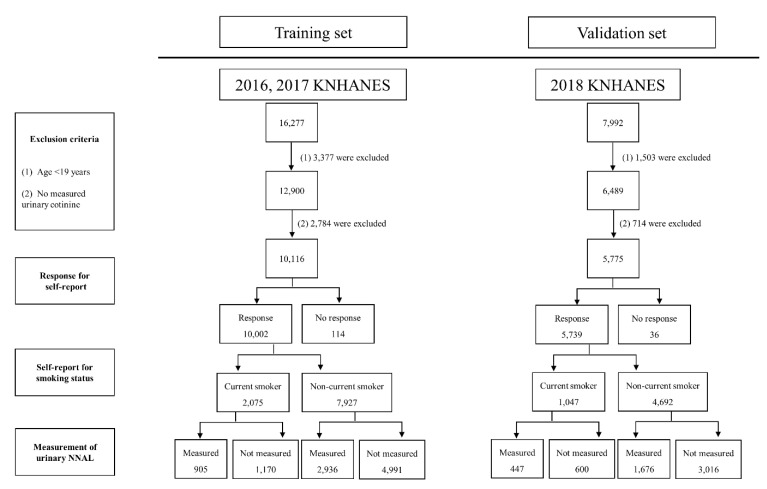
Schematic flow chart of the study design and exclusion criteria.

**Figure 2 ijerph-19-12147-f002:**
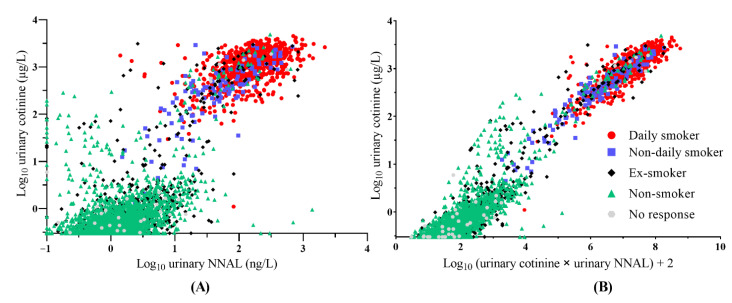
The scatter plots of log-transformed urinary cotinine and NNAL (**A**) and urinary cotinine∙NNAL (**B**) from total participants. The red, blue, black, green, and gray symbols represent daily smokers, non-daily smokers, ex-smokers, non-smokers, and participants without a response, respectively, in the training dataset. NNAL = 4-(methylnitrosamino)-1-(3-pyridyl)-1-butanol.

**Figure 3 ijerph-19-12147-f003:**
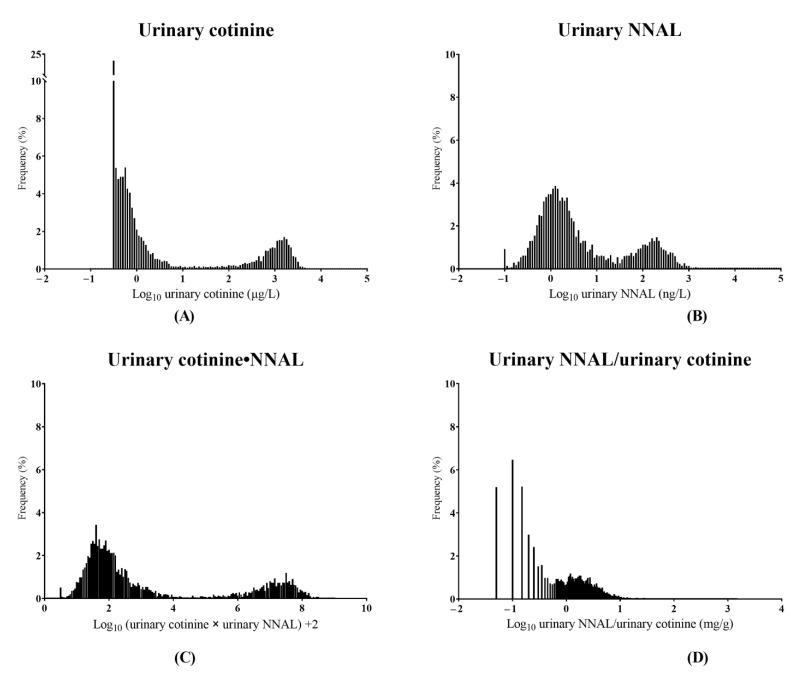
The distributions of four urinary biomarkers from non-smokers from total participants. Urinary cotinine (**A**), NNAL (**B**), cotinine∙NNAL (**C**), and NNAL/cotinine (**D**). NNAL = 4-(methylnitrosamino)-1-(3-pyridyl)-1-butanol.

**Figure 4 ijerph-19-12147-f004:**
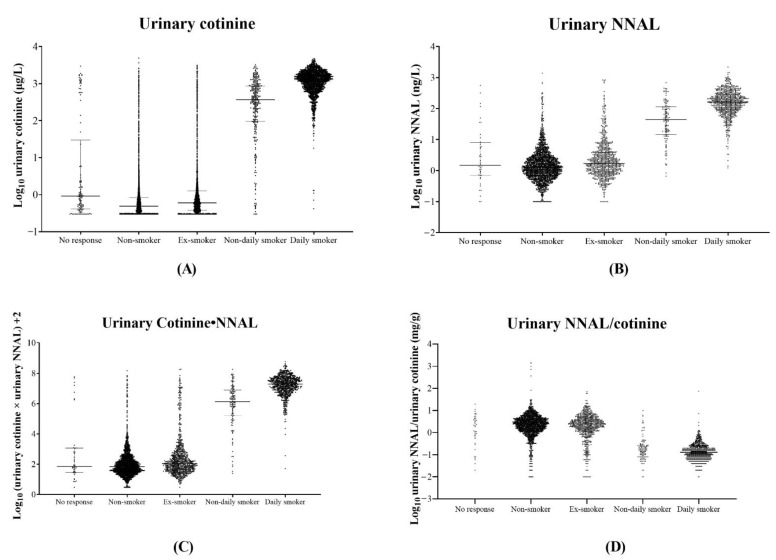
The scatter plots of four urinary biomarkers, according to smoking statuses. Urinary cotinine (**A**), NNAL (**B**), cotinine∙NNAL (**C**), and NNAL/cotinine (**D**). The horizontal solid lines represent the median values of each biomarker. NNAL 4-(methylnitrosamino)-1-(3-pyridyl)-1-butanol.

**Table 1 ijerph-19-12147-t001:** Characteristics of the study participants.

Data Characteristics	KNHANES (2016–2018)	
	Training Set	Validation Set
	2016, 2017	2018
Number	10,116	5775
Age, year (mean ± SD)	51.1 ± 16.5	49.1 ± 16.8
Sex, female (n, %)	5397 (53.5%)	3179 (55.0%)
Urinary cotinine (μg/L)	286.8 ± 648.0	300.0 ± 682.2
Subjects with measured NNAL concentration	3841 (37.9%)	2148 (37.2%)
Urinary NNAL (ng/L)	52.5 ± 131.7	45.9 ± 121.2
Response for self-report (n, %)	10,002 (98.9%)	5739 (99.4%)
Non-current smoker, self-report (n, %)	7927 (79.3%)	4692 (81.8%)
Non-smoker (n, %)	5803 (73.2%)	3428 (73.1%)
Ex-smoker (n, %)	2124 (26.8%)	1264 (26.9%)
Current smoker, self-report (n, %)	2075 (20.7%)	1047 (18.2%)
Daily smoker (n, %)	1796 (86.6%)	906 (86.5%)
Non-daily smoker (n, %)	279 (13.4%)	141 (13.5%)
Usage of nicotine replacement product (NRP)		
Current NRP user (n, %)	22 (0.2%)	10 (0.2%)
Non-current NRP user (n, %)	9980 (99.8%)	5729 (99.8%)
Usage of electronic cigarette (e-cigarette)		
Current e-cigarette user (n, %)	182 (1.8%)	163 (2.8%)
Non-current e-cigarette user (n, %)	9820 (98.2%)	5576 (97.2%)
Subjects with less or equal to LoD of urinary cotinine (n, %)	1225 (12.1%)	1216 (21.1%)
Subjects with less or equal to LoD of urinary NNAL (n, %)	9 (0.2%)	13 (0.6%)

NNAL = 4-(methylnitrosamino)-1-(3-pyridyl)-1-butanol, KNHANES = Korean National Health and Nutrition Examination Survey, LoD = the limit of detection.

**Table 2 ijerph-19-12147-t002:** The established optimal cut-off values of four urinary biomarkers using the training set and their diagnostic performance for smoking status classification.

Urinary Biomarker	Dataset	Optimal Cut-Off (95% CI)	Sensitivity (%)	Specificity (%)	Youden’s Index	AUC	95% CI	*p* Value
Cotinine (µg/L)	Training set	26.9	97.93	95.07	0.930	0.983	0.981–0.986	<0.0001
	Validation set	(9.3–33.9)	98.19	96.01		0.980	0.978–0.986	<0.0001
NNAL (ng/L)	Training set	12.3	96.24	94.18	0.904	0.980	0.975–0.984	<0.0001
	Validation set	(12.0–18.6)	96.87	92.72		0.975	0.971–0.984	<0.0001
Cotinine∙NNAL	Training set	4.0	98.45	95.33	0.938	0.985	0.981–0.989	<0.0001
	Validation set	(3.9–4.6)	98.66	94.62		0.987	0.980–0.991	<0.0001
NNAL/cotinine (mg/g)	Training set	0.6	94.70	88.15	0.828	0.937	0.929–0.945	<0.0001
	Validation set	(0.6–0.8)	96.20	89.35		0.942	0.930–0.952	<0.0001

The diagnostic performances of four urinary biomarkers were assessed using 3841 participants in the training set and 2148 participants in the validation set, whose urinary cotinine and NNAL were measured. NNAL = 4-(methylnitrosamino)-1-(3-pyridyl)-1-butanol.

**Table 3 ijerph-19-12147-t003:** A comparison of the diagnostic performance of four urinary biomarkers in e-cigarette users and NRP users.

Urinary Biomarker	Dataset	Optimal Cut-Off (95% CI)	NRP Users		E-Cigarette Users		Current Smokers	Non-Current Smokers
			Median (IQR)	Misclassification	Median (IQR)	Misclassification	Median (IQR)	Median (IQR)
Cotinine (µg/L)	Training set	26.9	1230 (433–2220)	81.8% (18/22)	1180 (650–2030)	2.7% (5/182)	1150 (608–1716)	0.54 (0.34–0.95)
	Validation set	(9.3–33.9)	767 (246–1676)	90.0% (9/10)	1332 (752.5–1884)	3.0% (5/163)	1312 (754–1900)	0.50 (0.31–0.85)
NNAL (ng/L)	Training set	12.3	93.4 (42.1–402)	90.0% (9/10)	112 (54.7–271)	10.1% (8/77) ^a^	149 (74.6–253)	1.39 (0.76–2.81)
	Validation set	(12.0–18.6)	N.T.	0/0	77.9 (16.8–195.3)	21.4% (15/70) ^b^	158 (68.1–273)	1.16 (0.67–2.25)
Cotinine∙NNAL	Training set	4.0	7.46 (6.78–7.55)	90.0% (9/10)	7.11 (6.66–7.59)	2.5% (2/77)	7.21 (6.71–7.61)	1.88 (1.47–2.39)
	Validation set	(3.9–4.6)	N.T.	0/0	6.98 (6.25–7.57)	2.9% (2/70) ^c^	7.28 (6.75–7.68)	1.78 (1.44–2.21)
NNAL/cotinine (mg/g)	Training set	0.6	0.19 (0.07–0.19)	90.0% (9/10)	0.13 (0.06–0.21)	6.3% (5/77)	0.14 (0.09–0.24)	2.55 (1.31–4.69)
	Validation set	(0.6–0.8)	N.T.	0/0	0.07 (0.03–0.16)	2.9% (2/70) ^c^	0.13 (0.07–0.22)	2.08 (1.13–3.53)

^a^. According to Fisher’s exact test, the misclassification rate of e-cigarette smokers between urinary cotinine and urinary NNAL showed a significant *p* value in the training set. (*p* = 0.024). ^b^. According to Fisher’s exact test, the misclassification rate of e-cigarette smokers between urinary cotinine and urinary NNAL showed a significant *p* value in the validation set. (*p* < 0.001). ^c^. According to Fisher’s exact test, the misclassification rate of e-cigarette smokers between urinary cotinine∙NNAL or NNAL/cotinine and urinary NNAL showed a significant *p* value in the validation set. (*p* = 0.001). NNAL = 4-(methylnitrosamino)-1-(3-pyridyl)-1-butanol, NRP = nicotine replacement product. N.T. = not tested.

**Table 4 ijerph-19-12147-t004:** A comparison of the diagnostic performance of four urinary biomarkers in non-current smokers, according to SHS exposure.

Urinary Biomarker	SHS Exposure		No SHS Exposure		Unclear SHS Exposure	
	Median	Less Than LoD	Median	Less Than LoD	Median	Less Than LoD
Cotinine (µg/L)	0.85 (0.59–1.70)	10.2% (33/320)	0.57 (0.40–0.94)	22.6% (956/4236)	1.26 (0.59–2.15)	3.0% (4/136)
NNAL (ng/L)	2.55 (1.32–4.99)	0.0% (0/49)	1.24 (0.72–2.38)	1.1% (13/1191)	3.02 (0.94–11.30)	0.0% (0/112)
Cotinine∙NNAL	2.40 (2.04–2.85)		1.87 (1.54–2.26)		2.55 (1.94–3.37)	
NNAL/cotinine (mg/g)	2.34 (1.20–4.01)		2.00 (1.08–3.45)		2.19 (1.16–4.52)	

SHS = secondhand smoke, NNAL = 4-(methylnitrosamino)-1-(3-pyridyl)-1-butanol, LoD = the limit of detection.

## Data Availability

The datasets used in this study were obtained from the Korean National Health and Nutrition Examination Survey (KNHANES) between 2016 and 2018.

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
