# Peer review of "Diagnostic Performance Evaluation of the Novel Index Combining Urinary Cotinine and 4-(Methylnitrosamino)-1-(3-pyridyl)-1-butanol in Smoking Status Verification and Usefulness for Trend Monitoring of Tobacco Smoking Exposure"

_ijerph, 2022, doi:10.3390/ijerph191912147_

Round 1
Reviewer 1 Report
Comments to the Author
The manuscript tried to prove that the novel
index, urinary cotinine∙NNAL, has a better diagnostic performance than the traditional
biomarkers, urinary cotinine or NNAL, for the classification of smoking status
in Korea. Based on the KNHANES
VII dataset in 2016-2018 (participants at ages 19+), the author applied the ROC curves and Youden’s index to
establish optimal cut-off values of urinary cotinine, NNAL, NNAL/cotinine, and
the novel index for smoking status classification. He/She also compared the diagnostic performance of
four urinary biomarkers in e-cigarette users, NRP users, and non-smokers by SHS
exposure status. The study
design and exclusion criteria are clear, and the issue this manuscript tried to solve is important.
However, the comparison of diagnostic performance for different
biomarkers is not sufficient to support the conclusion and the analysis of the results is not clear. I do have some comments and suggestions
that I think need some attention.

Reviewer 2 Report
Dear Authors,
The manuscript titled "Diagnostic performance evaluation of the novel index combining urinary cotinine and 4-(methylnitrosamino)-1-(3-pyridyl)-1-butanol in smoking status verification and trend monitoring of tobacco smoking exposure" is well written and clearly organized.
However, I have some questions and suggestions in order to improve the quality of your paper.
1- Table 1: Please, center the subtitles in the left columns.
2- What is the difference between non-current smokers and ex-smokers? Maybe I lost it in the main text.
3- In material and methods section, page 2 line 76, The Authors stated that 15,891 participants were enrolled in the study. However, in the flow chart (figure 1), the Authors reported that, considering both training and validating sets, they obtained no response from 150 participants and, in the end of the flow chart, the measurement of the urinary NNAL, was obtained only from 5964 (total) participants. Please explain and/or add the explanation on the main text.
4- Table 3: no data are present for validating set in the line of NNAL measurement in columns of NRP users, current smokers and non-current smokers.
Round 2
Reviewer 1 Report
Thanks for the response to my comments, and I think the manuscript is excellent now.